# Consistency Purification: Effective and Efficient Diffusion Purification towards Certified Robustness

**Yiquan Li**[1]* **Zhongzhu Chen**[2]* **Kun Jin**[2]* **Jiongxiao Wang**[1]* **Jiachen Lei**[3]
**Bo Li**[4] **Chaowei Xiao**[1]
[1]University of Wisconsin-Madison; [2] University of Michigan-Ann Arbor;
[3]California Institute of Technology ;[4]University of Illinois Urbana-Champaign

## Abstract

Diffusion Purification, purifying noised images with diffusion models, has been widely used for enhancing certified robustness via randomized smoothing. However, existing frameworks often grapple with the balance between efficiency and effectiveness. While the Denoising Diffusion Probabilistic Model (DDPM) offers an efficient single-step purification, it falls short in ensuring purified images reside on the data manifold. Conversely, the Stochastic Diffusion Model effectively places purified images on the data manifold but demands solving cumbersome stochastic differential equations, while its derivative, the Probability Flow Ordinary Differential Equation (PF-ODE), though solving simpler ordinary differential equations, still requires multiple computational steps. In this work, we demonstrated that an ideal purification pipeline should generate the purified images on the data manifold that are as much semantically aligned to the original images for effectiveness in one step for efficiency. Therefore, we introduced Consistency Purification, an efficiency-effectiveness Pareto superior purifier compared to the previous work. Consistency Purification employs the consistency model, a one-step generative model distilled from PF-ODE, thus can generate on-manifold purified images with a single network evaluation. However, the consistency model is designed not for purification thus it does not inherently ensure semantic alignment between purified and original images. To resolve this issue, we further refine it through Consistency Fine-tuning with LPIPS loss, which enables more aligned semantic meaning while keeping the purified images on data manifold. Our comprehensive experiments demonstrate that our Consistency Purification framework achieves state-of-the-art certified robustness and efficiency compared to baseline methods.

## 1 Introduction

Diffusion models were first proposed for high-quality image generation [1, 2, 3, 4, 5] and have been extended to generative tasks across various modalities, including audio [6, 7, 8], video [9, 10], and 3D object [11, 12, 13]. A diffusion model for image generation typically involves two key processes: (1) a forward diffusion process, which transforms the source image into an isotropic Gaussian by gradually adding Gaussian noise, and (2) the reverse diffusion process, which uses a Deep Neural Network (DNN) to perform iterative denoising starting from random Gaussian noise.

Due to the inherent denoising capability of diffusion models, there have been widely applied to improve the robustness of DNNs. This enhancement is achieved by Diffusion Purification [14, 15, 16, 17, 18], which purifies the network inputs to reduce the effects of various types of unforeseen corruptions or adversarial attacks. Among these, one particularly suitable and effective scenario of purification is to improve certified robustness through randomized smoothing [19] for image

---

*The first four authors contributed equally. Correspondence to: Jiongxiao Wang <jwang2929@wisc.edu>.

classification tasks. This method guarantees a tight robustness in the $\ell_2$ norm with a smoothed classifier. However, many previous works [19, 20, 21, 22, 23, 24] have shown that it still requires retraining with Gaussian augmented examples for each noise level to optimize the smoothed classifier. Diffusion models, capable of purifying Gaussian perturbed images before classification, can be seamlessly integrated with any base classifier to produce a smoothed classifier for arbitrary noise levels. This integration has been demonstrated to effectively enhance certified robustness, as supported by numerous studies [18, 25, 26, 27].

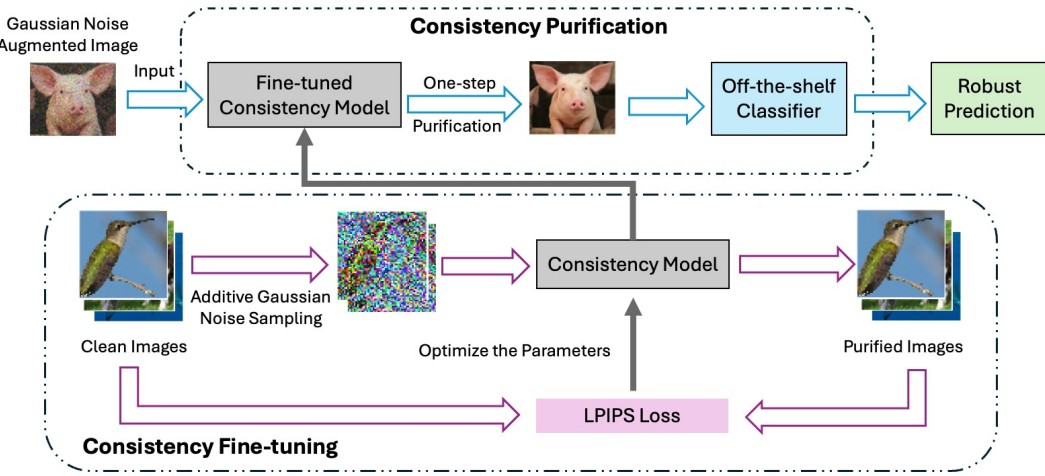

Figure 1: An illustration of Consistency Purification framework.

However, current diffusion purification for certified robustness via randomized smoothing still faces significant trade-offs between *efficiency* and *effectiveness*. Although Denoising Diffusion Probabilistic Model (DDPM) [28] only requires one single network evaluation in the purification process [25], it generates the mean of the posterior data distribution conditioned the noisy sample, which does not necessarily locate on the data manifold and may exhibit ambiguity during classification. To further improve diffusion purification, various methods such as DensePure [26], Local Smoothing [27] and Noised Diffusion Classifiers [29] are applied. However, these methods are considerably less efficient as they require multiple times of the computational costs compared to one-step DDPM. Another promising approach involves using the Probability Flow Ordinary Differential Equation (PF-ODE) [3]. It has offered a method to accelerate the sampling process [4] and achieved a closer distribution to the original data, well balancing efficiency and effectiveness. However, several computational steps are still needed to solve the ODE numerically.

To find a Pareto superior solution in terms of efficiency and effectiveness, we introduce a new framework, **Consistency Purification**, which integrating consistency models into diffusion purification with **Consistency Fine-tuning**. The consistency model is a novel category of diffusion models that learns the trajectory of the PF-ODE that transits the data distribution to the noisy distribution. It is trained to map any point along this trajectory back to its starting point. This property is desirable for diffusion purification, as it allows images with any scale of Gaussian noise to be directly purified to the clean images. Distilled from a pre-trained diffusion model by simulating the PF-ODE trajectory, the consistency model can generate high-quality in-distribution images in a single step, thereby ensuring both efficiency and effectiveness. However, since consistency models are primarily trained for image generation, it may not suffice to guarantee that the purified image that maintains the same semantic meaning as the original image. To address this issue, we propose adding a Consistency Fine-tuning step into the purification framework, which further fine-tunes the consistency model using Learned Perceptual Image Patch Similarity (LPIPS) [30] loss, aiming to minimize the perceptual differences between the purified and original images, thereby ensuring better semantic alignment, while at the same time, ensuring the purified images still lie on the data manifold.

We show that Consistency Purification is Pareto superior compared to baselines from two aspects. First of all, compared with effective methods like DensePure [26], Local Smoothing [27] and Noised Diffusion Classifiers [29], Consistency Purification is much more efficient since it enables single-step purification. Secondly, compared with efficient method like onestep-DDPM [25], we provide both

theoretical analysis and experiment results to support the effectiveness improvement of Consistency Purification. In Example 3.1, we show an one-dimensional example demonstrating that consistency model can generate on-manifold purified samples while onestep-DDPM does not have this property.

In Theorem 3.3, we show an important theoretical result that given a purifier, the lower the transport from the original distribution to the purified distribution (a measure of distance between probability distributions, see [31]), the higher the probability that the purified sample is sufficiently close to the original sample, and thus the better purification outcomes. Our experiment results verify that both the integration of consistency model in Consistency Purification and the further Consistency Fine-tuning decreases such transport and achieves better semantic alignment between purified samples and original samples.

Beyond the validation of our theory, we conduct comprehensive experiments to demonstrate the empirical improvements of Consistency Purification. Compared to various baseline settings, our approach has shown significant improvements, achieving an average 5% gain in performance over the previous onestep-DDPM under the same cost with single-step purification. These observations underscore our success in finding a Pareto superior diffusion purification framework in both efficiency and effectiveness for certified robustness.

## 2 Backgrounds

**Randomized Smoothing [19].** Randomized smoothing is designed to certify the robustness of a given classifier under $\ell_2$ norm perturbations. Given a base classifier $f$ and an input $\boldsymbol{x}$, randomized smoothing first defines the smoothed classifier by $g(\boldsymbol{x}) = \arg\max_c \mathbb{P}_{\boldsymbol{\epsilon} \sim \mathcal{N}(\boldsymbol{0}, \sigma^2 I)}(f(\boldsymbol{x} + \boldsymbol{\epsilon}) = c)$, where $\sigma$ is the noise level, which controls the trade-off between robustness and accuracy. [19] shows that $g(\boldsymbol{x})$ induces the certifiable robustness for $\boldsymbol{x}$ under the $\ell_2$ norm with radius $R$, where $R = \frac{\sigma}{2}\left(\Phi^{-1}(p_A) - \Phi^{-1}(p_B)\right)$, where $p_A$ and $p_B$ are the probability of the most probable class and "runner-up" class respectively; $\Phi$ is the inverse of the standard Gaussian CDF. The $p_A$ and $p_B$ can be estimated with arbitrarily high confidence via the Monte Carlo method.

**Continuous-Time Diffusion Model [3].** The diffusion model has two components: the *diffusion process* followed by the *reverse process*. Given an input random variable $\boldsymbol{x}_0 \sim p$, the diffusion process adds isotropic Gaussian noises to the data so that the diffused random variable at time $t$ is $\boldsymbol{x}_t = \sqrt{\alpha_t}(\boldsymbol{x}_0 + \boldsymbol{\epsilon}_t)$, s.t., $\boldsymbol{\epsilon}_t \sim \mathcal{N}(\boldsymbol{0}, \sigma_t^2 \boldsymbol{I})$, and $\sigma_t^2 = (1 - \alpha_t)/\alpha_t$, and we denote $\boldsymbol{x}_t \sim p_t$. The forward diffusion process can also be defined by the stochastic differential equation

$$\mathrm{d}\boldsymbol{x} = D(\boldsymbol{x}, t)\mathrm{d}t + G(t)\mathrm{d}\boldsymbol{w}, \tag{SDE}$$

where $\boldsymbol{x}_0 \sim p$, $D : \mathbb{R}^d \times \mathbb{R} \mapsto \mathbb{R}^d$ is the drift coefficient and typically has the form $D(\boldsymbol{x}, t) = D(t)\boldsymbol{x}$. $G : \mathbb{R} \mapsto \mathbb{R}$ is the diffusion coefficient, $\mathrm{d}t$ is an infinitesimal time step, and $\boldsymbol{w}(t) \in \mathbb{R}^n$ is the standard Wiener process.

The reverse process exists and removes the added noise by solving the reverse-time SDE [32]

$$\mathrm{d}\boldsymbol{x} = [D(t)\boldsymbol{x} - G(t)^2 \nabla_{\hat{\boldsymbol{x}}} \log p_t(\boldsymbol{x})]\mathrm{d}t + G(t)\mathrm{d}\overline{\boldsymbol{w}}, \tag{reverse-SDE}$$

where $p_t(\boldsymbol{x})$ denotes the marginal distribution at time $t$, and $\overline{\boldsymbol{w}}(t)$ is a reverse-time standard Wiener process. [3] defined the probability flow ODE (PF ODE) which has the same marginal distribution as reverse-SDE but can be solved much faster

$$\mathrm{d}\boldsymbol{x} = \left[D(t)\boldsymbol{x} - \tfrac{1}{2}G(t)^2 \nabla_{\boldsymbol{x}} \log p_t(\boldsymbol{x})\right] \mathrm{d}t. \tag{PF-ODE}$$

As shown in [4], the perturbation kernel of SDE has the general form

$$p_{0t}(\boldsymbol{x}(t) \mid \boldsymbol{x}(0)) = \mathcal{N}\left(\boldsymbol{x}(t); s(t)\boldsymbol{x}(0), s(t)^2 \sigma(t)^2 \mathbf{I}\right) \tag{perturbation-kernel}$$

where $s(t) = \exp\left(\int_0^t f(\xi)\mathrm{d}\xi\right)$ and $\sigma(t) = \sqrt{\int_0^t \frac{g(\xi)^2}{s(\xi)^2} \mathrm{d}\xi}$. Under this formulation, PF-ODE can written as

$$\mathrm{d}\boldsymbol{x} = \left[\frac{\dot{s}(t)}{s(t)}\boldsymbol{x} - s(t)^2 \dot{\sigma}(t)\sigma(t)\nabla_{\boldsymbol{x}} \log p\left(\frac{\boldsymbol{x}}{s(t)}; \sigma(t)\right)\right] \mathrm{d}t$$

where $\cdot$ denotes the time derivative and $p\left(\frac{\boldsymbol{x}}{s(t)}; \sigma(t)\right)$ denotes the marginal distribution at time $t$. In our context, we use the *EDM* parameter [4] where $s(t) = 1$ and $\sigma(t) = t$ which gives us a probability flow ODE

$$\mathrm{d}\boldsymbol{x} = -t\nabla_{\boldsymbol{x}} \log p_t(\boldsymbol{x})\mathrm{d}t. \tag{EDM-ODE}$$

We use $\{x_t\}_{t\in[0,1]}$ and $\{\hat{x}_t\}_{t\in[0,1]}$ to denote the diffusion process and the reverse process generated by SDE and reverse-SDE respectively, which follow the same distribution. We also use $\{\tilde{x}_t\}_{t\in[0,1]}$ to denote the reverse process generated by PF-ODE, which has the same marginal distribution as $\{x_t\}_{t\in[0,1]}$ and $\{\hat{x}_t\}_{t\in[0,1]}$ given $t$.

**Discrete-Time Diffusion Model (DDPM [28]).** DDPM constructs a discrete Markov chain $\{x_0, x_1, \cdots, x_i, \cdots, x_N\}$ as the forward process for the training data $x_0 \sim p$, such that $\mathbb{P}(x_i|x_{i-1}) = \mathcal{N}(x_i; \sqrt{1-\beta_i}x_{i-1}, \beta_i I)$, where $0 < \beta_1 < \beta_2 < \cdots < \beta_N < 1$ are predefined noise scales such that $x_N$ approximates the Gaussian white noise. Denote $\overline{\alpha}_i = \prod_{i=1}^{N}(1-\beta_i)$, we have $\mathbb{P}(x_i|x_0) = \mathcal{N}(x_i; \sqrt{\overline{\alpha}_i}x_0, (1-\overline{\alpha}_i)I)$, i.e., $x_t(x_0, \epsilon) = \sqrt{\overline{\alpha}_i}x_0 + (1-\overline{\alpha}_i)\epsilon, \epsilon \sim \mathcal{N}(0, I)$.

The reverse process of DDPM learns a reverse direction variational Markov chain $p_\theta(x_{i-1}|x_i) = \mathcal{N}(x_{i-1}; \mu_\theta(x_i, i), \Sigma_\theta(x_i, i))$. [28] defines $\epsilon_\theta$ as a function approximator to predict $\epsilon$ from $x_i$ such that $\mu_\theta(x_i, i) = \frac{1}{\sqrt{1-\beta_i}}\left(x_i - \frac{\beta_i}{\sqrt{1-\overline{\alpha}_i}}\epsilon_\theta(x_i, i)\right)$. Then the reverse time samples are generated by $\hat{x}_{i-1} = \frac{1}{\sqrt{1-\beta_i}}\left(\hat{x}_i - \frac{\beta_i}{\sqrt{1-\overline{\alpha}_i}}\epsilon_{\theta^*}(\hat{x}_i, i)\right) + \sqrt{\beta_i}\epsilon, \epsilon \sim \mathcal{N}(0, I)$, and the optimal parameters $\theta^*$ are obtained by solving $\theta^* := \arg\min_\theta \mathbb{E}_{x_0, \epsilon}\left[\|\epsilon - \epsilon_\theta(\sqrt{\overline{\alpha}_i}x_0 + (1-\overline{\alpha}_i), i)\|_2^2\right]$. [28] also provided a one-step approximate reconstruction of $x_0$ from any $x_t$,

$$x_0 \approx \hat{x}_0 = \left(x_t - \sqrt{1-\overline{\alpha}_t}\epsilon_\theta(x_t)\right)/\sqrt{\overline{\alpha}_t}. \qquad \text{(onestep-DDPM)}$$

**Consistency Model [33].** Given a solution trajectory of PF-ODE, the consistency model is defined as $D : (x_t, t) \mapsto x_\epsilon$. The model exhibits the property of self-consistency, ensuring that its outputs are consistent for arbitrary pairs of $(x_t, t)$ from the same PF-ODE trajectory; specifically, $D(x_t, t) = D(x_{t'}, t')$ for all $t, t' \in [\epsilon, T]$. As shown by the definition, consistency models are suitable for one-shot denoising, allowing for the recovery of $x_\epsilon$ from any noisy input $x_t$ in one network evaluation. Two distinct training strategies can be employed for training the consistency models: distillation mode and isolation mode. The primary distinction lies in whether the models distill the knowledge from pre-trained diffusion models or train from initial parameters. According to the experiments reported in [33], consistency models trained in the distillation mode have been shown to outperform those trained in isolation mode for generating high-quality images. Consequently, our paper only considers consistency models trained in the distillation mode.

## 3 Theoretical Analysis

In this section, we provide theoretical explanations on the advantages of Consistency Purification, with a focus on its purification performance improvement in terms of certified robustness over [25].

As demonstrated in [3], PF-ODE maintains the marginal distribution of reverse-SDE, thereby establishing a deterministic mapping between the noisy distribution $x_t$ and the data distribution $x_0$. In other words, PF-ODE guarantees that the purified sample lies on the data manifold, unlike onestep-DDPM, which lacks this assurance. We present here a simple one dimensional example for illustration.

**Example 3.1.** *Consider a one-dimensional space with a data set consisting of two samples $\{y_1, y_2\}$, where $y_1 = 1$ and $y_2 = -1$. The distribution can be represented as a mixture of Dirac delta distributions: $p_{data}(x) = \frac{1}{2}\left(\delta(x - y_1) + \delta(x - y_2)\right)$. By setting $s(t) = 1$ and $\sigma(t) = t$ in perturbation-kernel, the distribution at time $t$ becomes: $p_t(x) = \frac{1}{2t\sqrt{2\pi}}\left(e^{-\frac{1}{2}\left(\frac{x-1}{t}\right)^2} + e^{-\frac{1}{2}\left(\frac{x+1}{t}\right)^2}\right)$. Then*

$$
\begin{aligned}
\frac{\mathrm{d}\log p_t(x)}{\mathrm{d}x} &= \frac{-\left(\frac{x-1}{t}\right)e^{-\frac{1}{2}\left(\frac{x-1}{t^2}\right)^2} - \left(\frac{x+1}{t}\right)e^{-\frac{1}{2}\left(\frac{x+1}{t^2}\right)^2}}{2t\sqrt{2\pi}p_t(x)} \\
&= -\frac{x}{t^2} + \frac{e^{-\frac{1}{2}\left(\frac{x-1}{t}\right)^2} - e^{-\frac{1}{2}\left(\frac{x+1}{t}\right)^2}}{e^{-\frac{1}{2}\left(\frac{x-1}{t}\right)^2} + e^{-\frac{1}{2}\left(\frac{x+1}{t}\right)^2}}.
\end{aligned}
$$

*From the derivative formula $\frac{\mathrm{d}\log p_t(x)}{\mathrm{d}x}$, it's evident that $x = 0$ is an equilibrium point, and the right-hand side expression is Lipschitz continuous around $x = 0$ by L'Hôpital's rule. Thus, according to the Picard-Lindelöf theorem, any trajectory starting on either side of $x = 0$ will not cross this point. As PF-ODE drives $p_t(x)$ closer to the Dirac delta distribution $p_{data}(x)$ as $t$ approaches zero, any initial point on positive/negative side of $x = 0$ will eventually approach 1 or $-1$, i.e., the data manifold. Furthermore, in this example, PF-ODE generates not only a purified sample on the data*

*manifold but also closest to the noisy sample. This property is desirable as it establishes a relatively large "robust" neighborhood around each true data point, which implies high certified robustness and a significant certified radius, which will be further discussed later. With the consistency model, we do not need to solve the ODE but rather directly map the noisy sample to either $1/-1$ depending on its location relative to $x = 0$.*

*For comparison, given any $x$ and $t$, the onestep-DDPM will output a posterior mean that is*

$$\frac{e^{-\frac{1}{2}\left(\frac{x-1}{t}\right)^2} - e^{-\frac{1}{2}\left(\frac{x+1}{t}\right)^2}}{e^{-\frac{1}{2}\left(\frac{x-1}{t}\right)^2} + e^{-\frac{1}{2}\left(\frac{x+1}{t}\right)^2}} = \frac{e^{\frac{2x}{t^2}} - 1}{e^{\frac{2x}{t^2}} + 1}.$$

*The posterior mean will be near $1$ or $-1$ only when $t$ is sufficiently small compared to $\|x\|$. Otherwise, it deviates from the data manifold. In the case when $t$ is large, the posterior mean will be close to zero, locating in an ambiguous classification region. In adversarial purification [25, 26, 14], we typically select $t$ based on the variance of the noise added to the data sample rather than using an very small $t$. This practice helps avoid significant deviations in the posterior mean estimation due to the imperfect estimation of score/noise. With a very small $t$, even a slight bias in score/noise estimation can lead to a substantial deviation, resulting in a denoised sample even farther from the data manifold represented by $p_{data}(x)$.*

Additionally, PF-ODE is deterministic, eliminating the overhead of majority voting required when using reverse-SDE as a purifier [26]. The consistency model, which reduces ODE solving to a one-step mapping, further ensures purification has the same efficiency as onestep-DDPM while keeping the in-distribution property.

Though the consistency model enjoys both in-distribution property and one-step efficiency, it does not guarantee that the purified sample has the same semantic meaning as the original sample. This is because the derivation of PF-ODE only guarantees a mapping between noisy distribution and data distribution, which is sufficient for generation, but not enough for denoising purposes.

To address this concern, we first delineate the desired characteristics of the purifier. As evidenced in prior works [14, 25, 26, 34], an ideal purifier should yield a purified output situated within a proximate vicinity of the original input. It is generally presumed that such purified outputs retain the semantic meaning of the original inputs with a high probability. The disparity in semantic consistency between the noisy input and the purified output generated by PF-ODE arises due to the proximity of the purified output to other samples. In this regard, we propose quantifying this disparity through the notion of transport between the data distribution and the purified distribution, derived by introducing Gaussian perturbations to the data distribution and subsequently applying denoising via PF-ODE. Given an original sample $x$, Gaussian noise $\epsilon$, and purifier $d$, the mapping in the transport process is defined as $T : x \rightarrow d(x + \epsilon)$, which is probabilistic. We aim to demonstrate that a diminished transport between the data distribution and the purified distribution is conducive to a higher likelihood of the purified output being situated in proximity to the original sample, thereby preserving its semantic meaning.

We will leverage the following definition.

**Definition 3.2.** Given the data distribution $p$, Gaussian noise $\epsilon$, timestep $t$, and a purifier $d$, we define $\pi_t : x \rightarrow d(x + t\epsilon)$ and the "transport" under $g_t$ between the data distribution and purified distribution as $T_{\pi_t}(p) := \int \|x - \pi_t(x)\| \cdot p(x)dx$.

Intuitively, transport measures the distance between the original and purified samples, which should be small by an effective purifier. Below, we quantify this intuition and present our main theorem. See the detailed proof in Appendix B.

**Theorem 3.3.** *Given the transport $T_{\pi_t}(p)$ between the data distribution $p$ and the corresponding purified distribution under $g_t$, then for any $r > 0$, the probability that the distance between the original sample $x$ and purified sample $\hat{x} = \pi_t(x)$ is larger than $r$ is upper bounded by $\frac{T_{\pi_t}(p)}{r}$.*

Remark 3.4. By Theorem 3.3, the efficacy of the purifier hinges on two crucial factors: the transport $T_{\pi_t}(p)$ and the radius $r$. A theoretically perfect purifier would yield zero transport; however, this is unattainable due to the inherent randomness of $g_t$. Typically, we can optimize the parameter $t$ to minimize the transport, denoted as $T^* = \min_t \frac{T_{\pi_t}(p)}{r}$. In the context of classification tasks, the selection of $r$ also depends on the robustness of the classifier; a more robust classifier allows a larger $r$ to be chosen, thereby guarantee better purification efficacy.

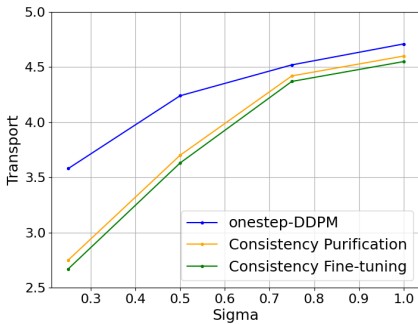

| | FID at different $\sigma$ | | |
| --- | --- | --- | --- |
| Loss | 0.25 | 0.5 | 1.0 |
| - - | 60.3 | 155.3 | 350.3 |
| $\ell_1$ | 96.8 | 205.7 | 383.6 |
| $\ell_2$ | 102.1 | 214.8 | 375.4 |
| LPIPS | **20.5** | **100.9** | **338.1** |

Figure 2: Transport between purified images and clean images with noise level $\sigma \in \{0.25, 0.5, 0.75, 1.0\}$.

Table 1: FID between purified and clean images on CIFAR-10 test set using different types of fine-tuning loss functions with noise level $\sigma \in \{0.25, 0.5, 1.0\}$.

For ensuring consistency in semantic meaning between the original and purified samples, it is insufficient merely to minimize their distance; it is also necessary that the purified sample resides on the data manifold, which is the in-distribution property we previously mentioned. To concurrently achieve both objectives, rather than solely focusing on minimizing the Euclidean distance between the original and purified samples, we opt to minimize the Learned Perceptual Image Patch Similarity (LPIPS) loss between them. This strategy aids in mitigating the risk of the purified sample deviating from the data manifold, thereby preserving semantic meaning. In Table 1, we show that using LPIPS is better than $\ell_1$ and $\ell_2$ loss for Consistency Fine-tuning when we want to guarantee the generated images are in-distribution, where lower FID scores indicate better in-distribution properties.

Figure 2 validates the effectiveness of Consistency Purification based on our results in Theorem 3.3, it shows that both the integration of consistency model in Consistency Purification and the further Consistency Fine-tuning can decrease the transport from the original distribution to the purified distribution. Specifically, we can see that Consistency Purification achieves a lower average distance from the purified sample to the original sample compared with onestep-DDPM, and Consistency Fine-tuning further decreases this average distance, indicating both components result in a lower transport and thus a better semantic alignment between purified samples and original samples.

## 4 Method

We propose our framework, Consistency Purification, with a further improved version using Consistency Fine-tuning.

### 4.1 Consistency Purification

We introduce Consistency Purification, directly applying consistency model as a purifier to integrate with a base classifier into smoothed classifier for randomized smoothing.

Following Diffusion Denoised Smoothing outlined in [25], it is necessary to establish a mapping between Gaussian noise augmented images required by randomized smoothing and the noised image in the ODE trajectory of consistency model. For a given consistency model purifier $D_\theta$, any noisy input $\boldsymbol{x}_t \sim \mathcal{N}(\boldsymbol{x}, t^2\boldsymbol{I})$ can be recovered to the trajectory's start $\boldsymbol{x}_\epsilon$ by directly passing it through the model with time $t$: $\boldsymbol{x}_\epsilon = D_\theta(\boldsymbol{x}_t, t)$.

When comparing this to the image augmented with additive Gaussian noise $\boldsymbol{x}_{rs} \sim \mathcal{N}(\boldsymbol{x}, \sigma^2\boldsymbol{I})$, which is required by randomized smoothing, we observe that $\boldsymbol{x}_{rs}$ and $\boldsymbol{x}_t$ share the same formula when $t = \sigma$. However, since the variances $\sigma \in \{\sigma_i\}_{i=1}^m$ may not be used during the training of the consistency model, we empirically select the nearest time step $t$ from the discrete time steps used in training for each $\sigma$.

For the entire time horizon $[\epsilon, T]$ with $N-1$ sub-interval boundaries $t_1 = \epsilon < t_2 < \cdots < t_N = T$, the time steps used in training are computed by: $t_i = (\epsilon^{1/\rho} + {}^{i-1}/_{N-1}(T^{1/\rho} - \epsilon^{1/\rho}))^\rho$, where $\rho = 7$.

Given the variance $\sigma$ of Gaussian noise used in randomized smoothing, we select the corresponding time step $t^*_\sigma$ for Consistency Purified Smoothing by $t^*_\sigma = \{t_i | \sigma \in (\frac{t_{i-1}+t_i}{2}, \frac{t_i+t_{i+1}}{2}]\}$.

## 4.2 Consistency Fine-tuning

To optimize the consistency model for aligning semantic meanings during purification, we fine-tune the purifier $D_\theta$ by minimizing the following loss function: $\mathcal{L}_\theta = \mathbb{E}\|\boldsymbol{x} - D_\theta(\boldsymbol{x}_\sigma, t^*_\sigma)\|_{\text{LPIPS}}$, where the expectation is taken with $\boldsymbol{x} \sim p_{data}$, $\sigma \sim \mathcal{U}\{\sigma_i\}^m_{i=1}$, $\boldsymbol{x}_\sigma \sim \mathcal{N}(\boldsymbol{x}, \sigma^2\boldsymbol{I})$. Here LPIPS denotes the distance computed by the Learned Perceptual Image Patch Similarity [30]. $p_{data}$ represents the distribution of the training data, from which clean images $\boldsymbol{x}$ are sampled. $\mathcal{U}\{\sigma_i\}^m_{i=1}$ denotes the uniform distribution over $m$ different noise scales $\sigma_i$ used for randomized smoothing. Typically, we select the scale set $\sigma_i \in \{0.25, 0.5, 1.0\}$, which is commonly used to compute the certified radius via randomized smoothing.

After obtaining the fine-tuned consistency model purifier $D_{\theta*}$, it can replace the original model used in Consistency Purified Smoothing to purify any noised image $\boldsymbol{x}_{rs}$ with Gaussian variance $\sigma_i$, resulting in the final purified image $\boldsymbol{x}_p$ by $\boldsymbol{x}_p = D_{\theta*}(\boldsymbol{x}_{rs}, t^*_{\sigma_i})$.

We present the detailed algorithm of our Consistency Purification in Appendix A.

# 5 Experiments

In this section, we begin by detailing the experimental settings, followed by our main results. Additionally, we conduct ablation studies to further demonstrate the effectiveness of our framework. All experiments are conducted with 1×NVIDIA RTX A5000 24GB GPU.

## 5.1 Experimental Settings.

**Dataset.** We evaluate the Consistency Purification framework on both CIFAR-10 [35] and ImageNet-64 [36]. CIFAR-10 contains $32 \times 32$ pixel images across 10 different categories while ImageNet-64 includes $64 \times 64$ pixel images across 1000 categories. Due to limited computational resources, we select 500 test images for CIFAR-10 from the 10,000 CIFAR-10 test set, choosing every 20th example in sequence (e.g., the 1st, 21st, 41st, etc.). Similarly, for the ImageNet-64 dataset, we sample 500 test examples from its 50,000 test examples using a fixed interval of 100.

**Consistency Purification.** For CIFAR-10, to demonstrate the effectiveness of Consistency Purification, we first perform purification with a public unconditional consistency model [37]. After that, to further improve the performance, we fine-tune the model with noise levels $\sigma$ sampling from $\{0.25, 0.5, 1.0\}$, shown as the (**+ Consistency Fine-tuning**). However, currently there is no publicly available unconditional consistency model checkpoint for the ImageNet dataset that can be used directly for purification purposes. The only available model is the conditional consistency model on ImageNet-64. Thus, here we trained an unconditional consistency model on ImageNet-64, initializing it with the existing conditional consistency model checkpoint. Details of the training process are included in Appendix C. Additionally, we also conduct Consistency Fine-tuning on ImageNet-64 model with noise levels $\sigma \in \{0.05, 0.15, 0.25\}$.

**Baselines.** For comparative analysis of CIFAR-10, we conduct baseline experiments under various settings. The first baseline involves onestep-DDPM, where we employ the 50-M unconditional improved diffusion models from [2] utilizing the one-shot denoising method [25] for purification. Given that our consistency model is distilled from an EDM model [4], we include EDM as our baselines, applying both one-shot denoising (onestep-EDM) and ODE solver (PF-ODE EDM) for purification. Additionally, we include the recent advancement in diffusion purification methods, Diffusion Calibration, as a baseline following [38], which fine-tunes the diffusion model with the guidance of classifier WideResNet28-10 to improve the purification accuracy under the specific classifier. While for ImageNet-64, due to the lack of public unconditional EDM model, we only include the comparison baseline with onestep-DDPM.

**Randomized Smoothing Settings.** We set $N = 10000$ for both CIFAR-10 and ImageNet as the number of sampling times used in randomized smoothing. We compute the certified radius for each test example at three different noise levels $\sigma \in \{0.25, 0.5, 1.0\}$ for CIFAR-10 and $\sigma \in \{0.05, 0.15, 0.25\}$ for ImageNet-64. Then we calculate the proportion of test examples whose radius

exceeds a specific threshold $\epsilon$. The highest accuracy among these noise levels is reported as the certified accuracy at $\epsilon$.

**Classifiers.** For the classifier used after purification for CIFAR-10, we employ ViT-B/16 model [39], which is pretrained on ImageNet-21k [36] and finetuned on CIFAR-10 dataset. In our ablation studies, we also use ResNet [40] and WideResNet [41] trained on CIFAR-10. For ImageNet-64, we make up-sampling on the 64×64 images and directly apply ViT-B/16 as the classifier.

## 5.2 Main Results.

We present the certified accuracy of Consistency Purification for both CIFAR-10 and ImageNet-64 dataset, with the results presented in Table 2. We also include the purification steps which decide whether the purifier needs multiple evaluation times through the networks (Multi Steps) other than a single network evaluation (One Step). As observed from Table 2, Consistency Purification significantly outperforms onestep-DDPM for both CIFAR-10 and ImageNet-64 with even higher certified accuracy with Consistency Fine-tuning. Besides, for CIFAR-10, the results also suggest the effectiveness of Consistency Purification with Consistency Fine-tuning when compared with more baseline methods such as onestep-EDM, PF-ODE EDM and Diffusion Calibration. For the detailed certified accuracy evaluation of fine-grained $\epsilon$ at different noise levels $\sigma$, we present the results in Figure 3 compared with the onestep-DDPM setting. All results have demonstrated that Consistency Purification is able to certify robustness with both efficiency and effectiveness.

Table 2: Certified Accuracy of Consistency Purification for CIFAR-10 and ImageNet-64.

| **CIFAR-10** | | Certified Accuracy at $\epsilon$ (%) | | | | |
|---|---|---|---|---|---|---|
| Method | Purification Steps | 0.0 | 0.25 | 0.5 | 0.75 | 1.0 |
| onestep-DDPM[25] | One Step | 87.6 | 73.6 | 55.6 | 39.2 | 29.6 |
| onestep-EDM | One Step | 87.4 | 76.2 | 58.8 | 40.8 | 32.4 |
| PF-ODE EDM | Multi Steps | 89.6 | 77.0 | 60.4 | 42.6 | 34.0 |
| Diffusion Calibration[38] | One Step | 90.2 | 76.4 | 57.2 | 42.6 | 32.4 |
| Consistency Purification | One Step | **90.4** | 77.2 | 59.8 | 42.8 | 33.2 |
| **+ Consistency Fine-tuning** | One Step | 90.2 | **79.4** | **62.4** | **43.8** | **35.4** |

| **ImageNet-64** | | Certified Accuracy at $\epsilon$ (%) | | | | |
|---|---|---|---|---|---|---|
| Method | Purification Steps | 0.0 | 0.05 | 0.15 | 0.25 | 0.35 |
| onestep-DDPM [25] | One Step | 55.2 | 44.8 | 33.4 | 15.2 | 8.8 |
| Consistency Purification | One Step | 62.4 | 54.2 | 35.2 | 19.8 | 13.0 |
| **+ Consistency Fine-tuning** | One Step | **68.6** | **58.0** | **37.4** | **23.4** | **17.4** |

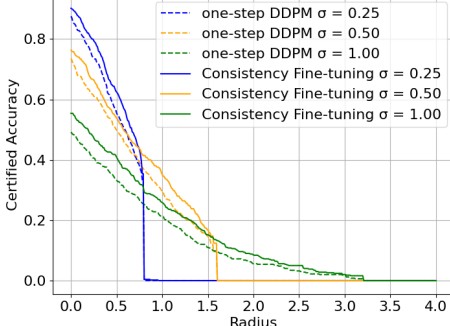 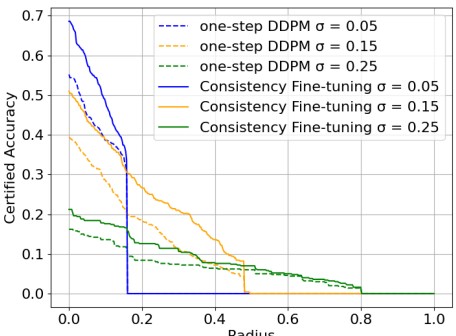

Figure 3: Certified Accuracy of Consistency Purification with fine-grained radius $\epsilon$. The left figure shows results on CIFAR-10, the right figure shows results on ImageNet-64. The lines demonstrate the certified accuracy of various radius $\epsilon$ under different Gaussian noise levels $\sigma$.

To better illustrate the significant improvement in certified robustness brought by Consistency Purification, we present visualizations of images after diffusion purification in Figure 4 for CIFAR-10 at a noise level of $\sigma = 0.5$, compared with the onestep-DDPM approach. As shown, our method produces significantly higher-quality purified images than onestep-DDPM. Furthermore, these purified images achieve a notably higher classification accuracy when evaluated by the same classifier. Additional visualization examples for ImageNet-64 are included in Appendix D.

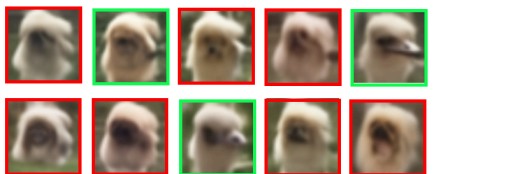    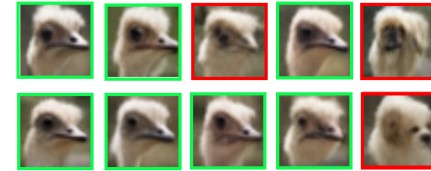

(a) Purified images by onestep-DDPM    (b) Purified images by Consistency Purification

Figure 4: Visualization of purified images after the diffusion purification by applying onestep-DDPM and Consistency Purification on CIFAR-10 with $\sigma = 0.5$ noise level. Identical noise patterns are applied to images at corresponding locations. A green border indicates that the purified image is correctly classified, while a red border denotes misclassification by the classifier.

## 5.3 Ablation Studies.

We conduct various ablation studies to evaluate the effectiveness of our proposed method.

**Comparison with Non-Diffusion-based Baselines.** To compare Consistency Purification with various non-diffusion-based approaches, we conducted additional experiments to compute the certified accuracy under three non-diffusion-based methods [19, 22, 42]. [19] first proposed training a classifier with noisy images to ensure certified robustness. Subsequent works [22, 42] build on [19]'s methodology, attempting to enhance the smoothed classifier by adding

|  | Certified Accuracy at $\epsilon$% | | | | |
| --- | --- | --- | --- | --- | --- |
| Methods | 0.0 | 0.25 | 0.5 | 0.75 | 1.0 |
| Randomized Smoothing [19] | 74.8 | 59.2 | 42.0 | 31.8 | 22.0 |
| Consistency Regularization [22] | 74.4 | 66.0 | 56.2 | 41.4 | 32.8 |
| Aces [42] | 74.6 | 66.4 | 57.0 | 43.6 | 32.8 |
| Consistency Purification | **90.4** | 77.2 | 59.8 | 42.8 | 33.2 |
| **+ Consistency Fine-tuning** | 90.2 | **79.4** | **62.4** | **43.8** | **35.4** |

Table 3: Certified Accuracy of Consistency Purification compared with non-diffusion-based baseline methods.

prediction consistency regularization, or incorporating per-sample bias. The experimental results presented in Tabel 3 show that our method surpasses all previous non-diffusion-based methods in achieving higher certified accuracy, particularly with a significantly high clean performance at $\epsilon = 0.0$. Furthermore, in contrast to non-diffusion-based methods, which incur significant costs by requiring additional fine-tuning of robust classifiers for each specific noise level, our method can be applied directly to any off-the-shelf classifiers, significantly broadening its practical applications.

**Fine-tuning Loss Functions.** To further demonstrate that LPIPS loss is the best choice considering both on-manifold purification and semantic meaning alignment, we assess the certified accuracy of Consistency Purification using different loss functions during Consistency Fine-tuning. Instead of LPIPS distance between the clean and purified images as the loss function, we experiment with $\ell_1$ and $\ell_2$ distances. Results in Table 4 indicate that Consistency Purification with LPIPS loss achieves the highest Certified Accuracy. In contrast, fine-tuning with $\ell_1$ and $\ell_2$ distances compromises the purification performance for certification. This demonstrates that fine-tuning with LPIPS loss function effectively aligns semantic meanings, whereas $\ell_1$ or $\ell_2$ distances may hurt them.

**Noise Levels Sampling Schedules during Consistency Fine-tuning.** In our experiments of Consistency Fine-tuning, we simply select the same sampling schedules of noise levels $\sigma \sim \mathcal{U}\{0.25, 0.5, 1.0\}$, uniformly sampling $\sigma$ used in randomized smoothing. To empirically demonstrate its effectiveness, we compare this approach with continuous sampling schedules where $\sigma \sim \mathcal{U}[0, 1]$. Results presented in Table 5 show that our discrete sampling schedule achieves higher certified accuracy. This indicates that fine-tuning with a discrete scale, aligned with the noise levels used in randomized smoothing, enhances certified robustness.

Table 4: Certified Accuracy of Consistency Purification with different loss functions during fine-tuning for CIFAR-10. "- -" represents the setting without fine-tuning.

|  | Certified Accuracy at $\epsilon\%$ | | | | |
|---|---|---|---|---|---|
| Distance | 0.0 | 0.25 | 0.5 | 0.75 | 1.0 |
| - - | **90.4** | 77.2 | 59.8 | 42.8 | 33.2 |
| $\ell_1$ | 89.4 | 76.4 | 59.6 | 42.4 | 31.4 |
| $\ell_2$ | 90.0 | 77.0 | 59.8 | 42.4 | 33.4 |
| LPIPS | 90.2 | **79.4** | **62.4** | **43.8** | **35.4** |

Table 5: Certified Accuracy of Consistency Purification with continuous and discrete sampling schedules during fine-tuning for CIFAR-10. "- -" represents the setting without fine-tuning.

|  | Certified Accuracy at $\epsilon\%$ | | | | |
|---|---|---|---|---|---|
| Schedules | 0.0 | 0.25 | 0.5 | 0.75 | 1.0 |
| - - | **90.4** | 77.2 | 59.8 | 42.8 | 33.2 |
| [0,1] | 89.0 | 76.2 | 59.8 | 43.2 | 33.8 |
| {0.25, 0.5, 1.0} | 90.2 | **79.4** | **62.4** | **43.8** | **35.4** |

**Generalizability with Different Classifiers.** We compute certified accuracy with various classifiers to test if our framework maintains its effectiveness with arbitrary classifiers. The results, presented in Table 6, compare Consistency Fine-tuning with Diffusion Calibration, an alternative method to fine-tune diffusion models for improving the certified robustness. When evaluated across different classifiers, including ViT-B/16, ResNet56, and WideNet28-10, our method outperforms Diffusion Calibration except certified accuracy at $\epsilon = 0.0$ on WRN28-10 model. It is worth noting that the Diffusion Calibration, which requires a specific classifier for guidance during fine-tuning, exhibits limitations, only achieving comparable performance with the guidance classifier WRN28-10. This demonstrates the advantages of Consistency Fine-tuning in generalizing across different classifiers.

**Fine-tuning Classifier vs. Fine-tuning Diffusion Model.** A potential concern with Consistency Fine-tuning is the higher certified accuracy and lower training cost associated with Fine-tuning the Classifier (CLS-FT) compared to our approach of Fine-tuning the Diffusion Model (DM-FT). However, our experiments, as shown in Table 7, indicate that DM-FT does not conflict with CLS-FT; rather, combining these two methods achieves even higher certified accuracy. On another hand, although CLS-FT yield slightly higher certified accuracy than DM-FT, its requirement for fine-tuning a specific classifier compromises the natural property of diffusion purification frameworks with arbitrary off-the-shelf classifiers, thus limiting the practical applicability.

Table 6: Certified Accuracy of Consistency Fine-tuning with different classifiers on CIFAR-10. The guidance classifier used in Diffusion Calibration is WideResNet28-10.

| | | Certified Accuracy at $\epsilon\%$ | | | | |
|---|---|---|---|---|---|---|
| Method | Classifier | 0.0 | 0.25 | 0.5 | 0.75 | 1.0 |
| | ViT-B/16 | 90.2 | 76.4 | 57.2 | 42.6 | 32.4 |
| Diffusion Calibration [38] | WRN28-10 | **88.2** | 76.4 | 59.2 | 42.0 | 31.8 |
| | ResNet56 | 86.0 | 72.8 | 52.6 | 35.2 | 25.8 |
| | ViT-B/16 | 90.2 | **79.4** | **62.4** | **43.8** | **35.4** |
| **Consistency Fine-tuning** | WRN28-10 | 88.0 | 76.4 | 59.8 | 42.8 | 33.0 |
| | ResNet56 | 87.2 | 74.8 | 57.6 | 38.2 | 30.2 |

Table 7: Certified Accuracy of Fine-tuning the Diffusion Model (DM-FT) compared with Fine-tuning the Classifier (CLS-FT) in diffusion purification frameworks on CIFAR-10.

| | | Certified Accuracy at $\epsilon\%$ | | | | |
|---|---|---|---|---|---|---|
| DM-FT | CLS-FT | 0.0 | 0.25 | 0.5 | 0.75 | 1.0 |
| - | - | 90.4 | 77.2 | 59.8 | 42.8 | 33.2 |
| ✓ | - | 90.2 | 79.4 | 62.4 | 43.8 | 35.2 |
| - | ✓ | 90.4 | 79.8 | 63.4 | 44.2 | 35.2 |
| ✓ | ✓ | **90.8** | **80.0** | **64.8** | **44.6** | **36.8** |

# 6 Conclusion

In this paper, we introduced Consistency Purification, a novel framework proposed to enhance certified robustness via randomized smoothing. By incorporating consistency models into diffusion purification approach and further refining them through Consistency Fine-tuning, our empirical experiments have demonstrate the framework's ability to achieve high certified robustness efficiently with one single network evaluation for purification.

**Limitations.** A notable limitation of our study is that our empirical results do not include computing certified robustness of high-resolution images such as ImageNet 256×256. This constraint is due to the absence of publicly available checkpoints for the consistency model at this resolution. Additionally, training a consistency model for ImageNet 256×256 would require huge computing resources, which are currently beyond our affordability. However, our framework is designed for adaptability and could be easily extended to ImageNet 256×256 once these checkpoints become available. As a result, our empirical evaluations in this paper are limited to the CIFAR-10 and ImageNet 64×64 datasets.

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

# A    Consistency Purification Algorithm

We provide detailed descriptions of Consistency Purification in the following algorithms. Algorithm 1 presents the function of Consistency Fine-tuning and Consistency Purification respectively. Algorithm 2 shows the randomized smoothing algorithm from [19] with applying Consistency Purification to do prediction and compute the certified radius.

---

**Algorithm 1** Consistency Fine-tuning and Consistency Purification

---

**Input:** Consistency model purifier $D_\theta$ where $\theta$ represents the model parameters. Noise levels used in randomized smoothing $\{\sigma_i\}_{i=1}^m$. Arbitrary classification model $f_{\text{clf}}$. Fine-tuning learning rate $\eta$.

1: **function** CONSISTENCYFINE-TUNING($D_\theta$)
2:     **repeat**
3:         **sample** $x \in$ Training Dataset, $\sigma \in \{\sigma_i\}_{i=1}^m$
4:         $x_\sigma \leftarrow x + \mathcal{N}(0, \sigma^2 \boldsymbol{I})$
5:         $t_\sigma^* \leftarrow$ GETTIMESTEP($\sigma$)
6:         $\mathcal{L} \leftarrow$ LPIPS($x, D_\theta(x_\sigma, t^*)$)
7:         $\theta \leftarrow \theta - \eta \nabla_\theta \mathcal{L}$
8:     **until** convergence
9:     **return** $D_\theta$
10: **end function**

11: **function** CONSISTENCYPURIFICATION($f_{\text{clf}}, x, \sigma$)
12:     $t_\sigma^* \leftarrow$ GETTIMESTEP($\sigma$)
13:     $\boldsymbol{x}_{rs} \leftarrow \boldsymbol{x} + \mathcal{N}(0, \sigma^2 I)$
14:     $\boldsymbol{x}_p \leftarrow D_{\theta^*}(\boldsymbol{x}_{rs}, t_\sigma^*)$
15:     $y \leftarrow f_{\text{clf}}(\boldsymbol{x}_p)$
16:     **return** $y$
17: **end function**

18: **function** GETTIMESTEP($\sigma$)
19:     $t_i \leftarrow (\epsilon^{1/\rho} + \frac{i-1}{N-1}(T^{1/\rho} - \epsilon^{1/\rho}))^\rho$ for $i \in \{1, \ldots, N\}$
20:     $t_\sigma^* \leftarrow$ find $\{t_i | \sigma \in \left( \frac{t_{i-1}+t_i}{2}, \frac{t_i+t_{i+1}}{2} \right] \}$
21:     **return** $t_\sigma^*$
22: **end function**

---

# B    Proof of Theorem 3.3

**Theorem 3.3.** *Given the transport $T_{\pi_t}(p)$ between the data distribution $p$ and the corresponding purified distribution under $g_t$, then for any $r > 0$, the probability that the distance between the original sample $\boldsymbol{x}$ and purified sample $\hat{\boldsymbol{x}} = \pi_t(\boldsymbol{x})$ is larger than $r$ is upper bounded by $\frac{T_{\pi_t}(p)}{r}$.*

*Proof.* We can leverage the Markov's inequality. Because

$$
\begin{aligned}
\mathbb{E}[\|\boldsymbol{x} - \hat{\boldsymbol{x}}\|] &= \int_{\|\boldsymbol{x}-\hat{\boldsymbol{x}}\| \leq r} \|\boldsymbol{x} - \hat{\boldsymbol{x}}\| \cdot p(\boldsymbol{x})d\boldsymbol{x} + \int_{\|\boldsymbol{x}-\hat{\boldsymbol{x}}\| > r} \|\boldsymbol{x} - \hat{\boldsymbol{x}}\| \cdot p(\boldsymbol{x})d\boldsymbol{x} \\
&\geq \int_{\|\boldsymbol{x}-\hat{\boldsymbol{x}}\| > r} \|\boldsymbol{x} - \hat{\boldsymbol{x}}\| \cdot p(\boldsymbol{x})d\boldsymbol{x} \\
&\geq \int_{\|\boldsymbol{x}-\hat{\boldsymbol{x}}\| > r} r \cdot p(\boldsymbol{x})d\boldsymbol{x} \\
&= r \cdot P(\|\boldsymbol{x} - \hat{\boldsymbol{x}}\| > r),
\end{aligned}
$$

we have

$$P(\|\boldsymbol{x} - \hat{\boldsymbol{x}}\| > r) \leq \frac{\mathbb{E}[\|\boldsymbol{x} - \hat{\boldsymbol{x}}\|]}{r}$$
$$= \frac{\mathbb{E}[\|\boldsymbol{x} - \pi_t(\boldsymbol{x})\|]}{r}$$
$$= \frac{T_{\pi_t}(p)}{r}.$$

$\square$

---

**Algorithm 2** Randomized Smoothing [19]

---

**Input:** Sampling times for prediction $n$. Sampling times for certification $N$. Significant confidence level $\alpha$. Function LOWERCONFBOUND($k, n, 1 - \alpha$) returns a one-sided (1-$\alpha$) lower confidence interval for the Binomial parameter $p$ given that $k \sim \text{Binomial}(n, p)$.

1: **function** PREDICT($f_{\text{clf}}, \boldsymbol{x}, \sigma, n, \alpha$)
2:     counts $\leftarrow 0$
3:     **for** $i \in \{1, 2, \ldots, n\}$ **do**
4:         $y \leftarrow$ CONSISTENCYPURIFICATION($f_{\text{clf}}, \boldsymbol{x}, \sigma$)
5:         counts[y] $\leftarrow$ counts[y] + 1
6:     **end for**
7:     $\hat{y}_A, \hat{y}_B \leftarrow$ top two labels in counts
8:     $n_A, n_B \leftarrow$ counts[$\hat{y}_A$],counts[$\hat{y}_B$]
9:     **if** BINOMTEST($n_A, n_A + n_B, \frac{1}{2}$) $\leq \alpha$ **then**
10:         **return** $\hat{y}_A$
11:     **else**
12:         **return** Abstain
13:     **end if**
14: **end function**
15:
16: **function** CERTIFY($f_{\text{clf}}, \boldsymbol{x}, \sigma, n, N, \alpha$)
17:     counts0 $\leftarrow 0$
18:     **for** $i \in \{1, 2, \ldots, n\}$ **do**
19:         $y \leftarrow$ CONSISTENCYPURIFICATION($f_{\text{clf}}, \boldsymbol{x}, \sigma$)
20:         counts0[y] $\leftarrow$ counts0[y] + 1
21:     **end for**
22:     $\hat{y}_A \leftarrow$ top label in counts0
23:     counts $\leftarrow 0$
24:     **for** $i \in \{1, 2, \ldots, N\}$ **do**
25:         $y \leftarrow$ CONSISTENCYPURIFICATION($f_{\text{clf}}, \boldsymbol{x}, \sigma$)
26:         counts[y] $\leftarrow$ counts[y] + 1
27:     **end for**
28:     $\underline{p_A} \leftarrow$ LOWERCONFBOUND(counts[$\hat{y}_A$], $N, 1 - \alpha$)
29:     **if** $\underline{p_A} > \frac{1}{2}$ **then**
30:         **return** prediction $\hat{y}_A$ and radius $\sigma \Phi^{-1}(\underline{p_A})$
31:     **else**
32:         **return** Abstain
33:     **end if**
34: **end function**

---

## C   Training Unconditional Consistency Model for ImageNet-64

We train an unconditional consistency model for ImageNet-64 from the public available conditional version by transiting the class embedding layers to a learnable token, initialization with average class embeddings. For each model forwarding, this token will be combined with the time embeddings for computation. After that, we train the conditional consistency model, initialized with the unconditional model's parameters, on ImageNet-64 training set for 120k steps.

# D  Purified Images Visualization for ImageNet-64

We present visualization images after diffusion purification by applying onestep-DDPM and Consistency Purification for ImageNet-64 under the noise level $\sigma = 0.25$ in Figure 5.

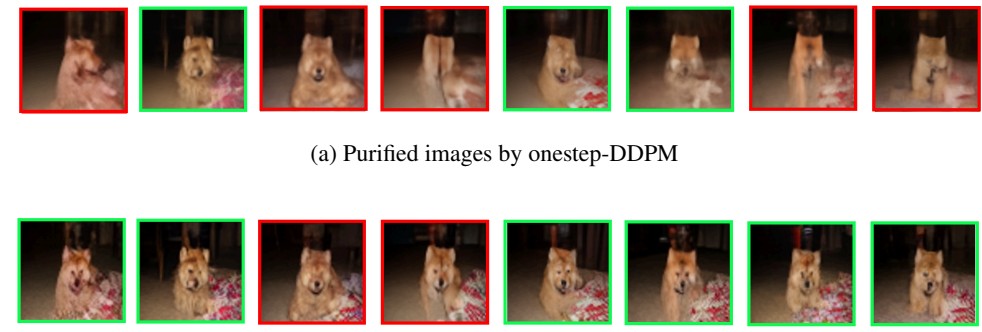

(a) Purified images by onestep-DDPM

(b) Purified images by Consistency Purification

Figure 5: Visualization of purified images after the diffusion purification by applying onestep-DDPM and Consistency Purification on ImageNet-64 with $\sigma = 0.25$ noise level. Identical noise patterns are applied to images at corresponding locations. A green border indicates that the purified image is correctly classified, while a red border denotes misclassification by the classifier.

