# OpenReview forum: "Consistency Purification: Effective and Efficient Diffusion Purification towards Certified Robustness"
_NeurIPS.cc/2024/Conference — NeurIPS 2024 poster_

### Official Review · Reviewer_aFWW · 2024-06-26

**Soundness:** 3
**Presentation:** 3
**Contribution:** 3
**Rating:** 5
**Confidence:** 3

**Summary:**

This paper presents an innovative approach to image purification using diffusion models, known as Consistency Purification. Traditional methods like the Denoising Diffusion Probabilistic Model (DDPM) and the Stochastic Diffusion Model face challenges in balancing efficiency and effectiveness. While DDPM provides efficient purification, it fails to ensure that purified images lie on the data manifold. On the other hand, the Stochastic Diffusion Model successfully places images on the data manifold but is computationally intensive. The proposed Consistency Purification leverages a consistency model distilled from the Probability Flow Ordinary Differential Equation (PF-ODE), achieving on-manifold purification in a single step. This approach is further refined through Consistency Fine-tuning with LPIPS loss, enhancing semantic alignment between purified and original images. Comprehensive experiments demonstrate that this framework achieves state-of-the-art certified robustness and efficiency compared to baseline methods.

**Strengths:**

1. Introducing the consistency model to the purification domain is an interesting idea, because it utilizes the consistency model's feature that maps noised images back to origin. , and experimental results show significant improvements in classification efficiency.
2. The writing and logical flow are clear and well-structured.

**Weaknesses:**

1. There is no comparison with multi-step DDPM methods. Providing these results and the corresponding speedup would make the work more solid.
2. The impact of noise intensity on experimental results is not mentioned. Could you please show this in the rebuttal phase?
3. It is unclear how the model performs if the noise is not Gaussian. Providing results for different noise distributions would enhance the robustness and applicability of the method.
4. Including visualizations of the purified images before and after the purification process would make the results more intuitive and compelling.

**Questions:**

refer to weakness

**Limitations:**

refer to weakness

---

> ### Author Rebuttal · Authors · 2024-08-07
>
> > Question A: Comparison with multistep DDPM methods.
>
> We conducted an experiment with multistep DDPM using 25 sampling steps, and the results are shown in the table below. We found that multistep DDPM has lower certified accuracy than onestep DDPM. This aligns with Carlini et al.'s [1] finding that multistep DDPM sampling adds unwanted details to the image, altering its semantic meaning. Our method provides a higher certified radius and is 25 times faster than multistep DDPM.
>
> |                           | 0.0  | 0.25 | 0.50 | 0.75 | 1.00 |
> |---------------------------|------|------|------|------|------|
> | onestep-DDPM | 87.6 | 73.6 | 55.6 | 39.2 | 39.6 |
> | multistep-DDPM | 86.2 | 72.4 | 55.8 | 40.6 | 31.8 |
> | Consistency Purification | **90.4** | 77.2 | 59.8 | 42.8 | 33.2 |
> | + Consistency Fine-tuning | 90.2 | **79.4** | **62.4** | **43.8** | **35.4** |
>
>
> > Question B: Impact of noise intensity on experimental results.
>
> In all experiments presented in our paper, we compute the certified radius for each test example at three distinct noise intensities, with $\sigma \in {0.25, 0.5, 1.00}$ for CIFAR-10 and $\sigma \in {0.05, 0.15, 0.25}$ for ImageNet-64. We then calculate the proportion of test examples whose radius exceeds a specific threshold $\epsilon$. The highest accuracy among these noise intensities is reported as the certified accuracy at $\epsilon$. For more detailed results of the certified accuracy under various noise intensities, we have included the results for each $\sigma$ in the tables below. The results clearly show that our method consistently outperforms other methods across all noise levels.
>
> Certified Accuray under various $\sigma$ for CIFAR-10
>
> | Method                    | $\sigma$ | 0.00 | 0.25 | 0.50 | 0.75 | 1.00 |
> |---------------------------|------|------|------|------|------|------|
>  | onestep-DDPM          | 0.25 | 87.6 | 73.6 | 55.6 | 37.8 | 0.0  |
>  |                           | 0.5  | 73.6 | 61.0 | 49.8 | 39.2 | 29.6 |
> |                           | 1.0  | 49.2 | 40.6 | 33.2 | 26.6 | 21.4 |
>  |Consistency Purification | 0.25 | 90.4 | 77.2 | 59.8 | 37.8 | 0.0  |
> |                           | 0.5  | 77.2 | 65.0 | 52.6 | 42.8 | 33.2 |
> |                           | 1.0  | 52.0 | 44.2 | 36.4 | 29.4 | 23.8 |
> | + Consistency Fine-tuning           | 0.25 | **90.2** | **79.4** | **62.4** | **43.2** | 0.0  |
> |                           | 0.5  |**76.4** | **66.4** | **53.8** | **43.8** | **35.4** |
> |                           | 1.0  | **55.4** | **47.2** | **41.0** | **32.0** | **26.0** |
>
> Certified Accuray under various $\sigma$ for ImageNet-64
>
> | Method                    | $\sigma$ | 0.00 | 0.05 | 0.15 | 0.25 | 0.35 |
> |---------------------------|------|------|------|------|------|------|
> | onestep-DDPM         | 0.05 | 55.2 | 44.8 | 33.4 | 0.0  | 0.0  |
> |                           | 0.15 | 38.2 | 35.4 | 21.8 | 15.2 | 8.8  |
> |                           | 0.25 | 13.2 | 10.8 | 8.4  | 5.4  | 4.0  |
> | Consistency Purification| 0.05 | 62.4 | 54.2 | 35.2 | 0.0  | 0.0  |
> |                           | 0.15 | 41.8 | 37.2 | 25.4 | 19.8 | 13.0 |
> |                           | 0.25 | 16.2 | 13.8 | 13.0 | 6.2  | 5.8  |
> | + Consistency Fine-tuning         | 0.05 | **68.6** | **58.0** | **37.4** | 0.0  | 0.0  |
> |                           | 0.15 | **51.0** | **43.6** | **32.4** | **23.4** | **17.4** |
> |                           | 0.25 | **18.2** | **15.4** | **13.2** | **8.0**  | **7.2**  |
>
>
> > Question C: Model performances under non-Gaussian noise.
>
> In the scheme of randomized smoothing, the noise added to the image is chosen to be Gaussian for the simplicity of certified radius calculation, which is the standard practice in previous work [1, 2, 3]. We admit that it will be an interesting direction to conduct non-Gaussian noise to the randomized smoothing approach in future work.
>
> > Question D:  Visualizations of the purified images before and after the purification process
>
> Please refer to the visualization of images before and after purification via the following anonymous link: https://anonymous.4open.science/r/Consistency-Purification-9B5F/README.md. We have included purified images for CIFAR-10 at a noise level of 0.5 and for ImageNet-64 at a noise level of 0.25. For both datasets, our method produces images that are more detailed and accurate than those processed by One-Step DDPM [1]. This enhancement in image quality partially explains why our method achieves significantly higher certified accuracy compared to the One-Step DDPM method.
>
> [1] Nicholas Carlini et al. (certified!!) adversarial robustness for free! arXiv preprint arXiv:2206.10550, 2022.
>
> [2] Nie, Weili, et al. Diffusion models for adversarial purification. arXiv preprint arXiv:2205.07460 (2022).
>
> [3] Xiao, Chaowei, et al. Densepure: Understanding diffusion models towards adversarial robustness. arXiv preprint arXiv:2211.00322 (2022).

---

> > ### Author Response · Authors · 2024-08-11
> > **Look forward to your reply**
> >
> > Dear Reviewer aFWW,
> >
> > The deadline for the discussion period is approaching. We have provided our rebuttal material and hopefully could address your concerns. Your feedback is highly valuable to us, and we would greatly appreciate it if you could take some time to review our response.
> >
> > Best Regards,
> >
> > Authors

---

> > > ### Comment · Reviewer_43Xy · 2024-08-14
> > >
> > > I understand the authors’ response and efforts, i keep my rating.

---

### Official Review · Reviewer_wtha · 2024-07-11

**Soundness:** 3
**Presentation:** 3
**Contribution:** 2
**Rating:** 5
**Confidence:** 4

**Summary:**

The paper introduces the Consistency Model for one-step purification, ensuring the data manifold of the purified samples while maintaining the efficiency of one-step purification. At the same time, the paper proposes Consistency Finetuning to fine tune the Consistency Model to ensure semantic consistency of the purified samples.

**Strengths:**

1. The paper achieves a trade-off between efficiency and effectiveness, realizing Consistency Purification in a one-step manner.
2. The author conductes both theoretical and experimental verification of the proposed method.
3. The paper achieves consistent performance improvement in the experiment.

**Weaknesses:**

1. The paper's innovation is limited, using existing Consistency Models and LPIPS loss to form the diffpure framework.
2. There are some descriptions not easy to understand. Suggest the author to provide a brief explanation and clarification when transport (Page 3, line 77) first appears in the Introduction. Does Remark 3.4 (line202) need to be bolded?
3. The paper's validation set on ImageNet-64 is too small, and 100 images are difficult to accurately reflect its validity. Meanwhile, one of the purposes of this paper is to enhance the effectiveness, but the experimental dataset does not align with this original intention.

**Questions:**

1. In Example 3.1, the author states in lines 177-178 that if distribution consistency is already being ensured, it should already be a relatively strong constraint. Why can only ensure generation and not enough for denoising?
2. The paper's statements in 188-191 and the proof in Theorem 3.3 demonstrate the importance of small transport. However, the ideal denoising should be to transform d into x. For a defined transport, it is natural and reasonable to require it to be small. What the author needs to prove more is whether this Gaussian transport lower can be transferred to adversarial attacks, rather than just proving that lower is good?

**Limitations:**

The paper has addressed the limitations.

---

> ### Author Rebuttal · Authors · 2024-08-07
>
> > Concerns about the limited innovation in our paper
>
> We summarize our contributions and novelty in three points:
> 1. Using Consistency Models(CM) are efficient and effective:
>
> We provide theoretical (section 3, lines 143-174) and empirical support (Table 2) showing that CM enables us to achieve a Pareto-optimal purification framework compared to previous frameworks. First, CM significantly improves purification efficiency through its one-step property, compared to multi-step models like DDPM [1], Score-SDE [2], and EDM [3]. Second, CM produces on-manifold purified samples, enhancing purification effectiveness over other one-step models, such as one-step DDPM [1, 4], which can generate off-manifold samples that have semantic ambiguity (leading to low classifier confidence or misclassification). Third, compared to non-diffusion-based models, we can leverage off-the-shelf CM without requiring additional training, such as adversarial classifiers.
>
> 2. Leveraging CM alone is insufficient:
>
> We provide theoretical (section 3, lines 175-207) and empirical results (Table 1, Figure 2) to show this and further consistency-finetuning with LPIPS loss is necessary. The original CM cannot guarantee semantic alignment and can potentially recover a noisy bookstore image as a market image. Our consistency-finetuning aims to mitigate this issue.
>
> 3. Why LPIPS instead of L1 & L2?
>
> We provide theoretical support (section 3, lines 208-216) and experimental results (Table 1) to explain why typical L1 and L2 losses do not work and why LPIPS loss is appropriate for consistency fine-tuning. In particular, since the LPIPS loss measures the semantic difference rather than just the distance between samples, it ensures correct classification without requiring the purified sample to be identical to the original image. Conversely, minimizing L1 or L2 loss could potentially disrupt the CM structure and lead to off-manifold samples and semantic ambiguity.
> [1] Ho et al. Denoising diffusion probabilistic models. NeurIPS, 2020.
> [2] Song et al. Score-based generative modeling through stochastic differential equations. ICLR, 2021.
> [3] Karras et al. Elucidating the design space of diffusion-based generative models. NeurIPS, 2022.
> [4] Carlini & Tramer et al.(Certified!!) Adversarial Robustness for Free! ICLR, 2023.
>
> > Brief explanation and clarification of the transport definition
>
> Additionally, our transport (Definition 3.2) aligns with the standard definition in optimal transport theory [1]. Remark 3.4 suggests if the recovered sample distribution is closer to the original and the classifier is more robust, purification performance improves, we can bold it in the final version.
>
> [1] Peyré et al. Computational optimal transport: With applications to data science.
>
> > Small validation set on ImageNet-64
>
> To accurately evaluate the certified radius for our experiments on ImageNet-64, we expanded our test dataset to include 500 samples. We also present the variance across five subsets of 100 sample results. The results, shown in the following table, illustrate that consistency purification with consistency fine-tuning continues to achieve the highest certified accuracy. Additionally, the small variance observed across the five subsets of 100 samples further demonstrates the validity of our evaluation.
>
> |  | 0.0  | 0.05 | 0.15 | 0.25 | 0.35 |
> |-------------|------|------|------|------|------|
> | onestep-DDPM | 55.2±1.17 | 44.8±0.40 | 33.4±0.75 | 15.2±0.37 | 8.8±0.75 |
> | Consistency Purification | 62.4±1.02 | 54.2±1.16 | 35.2±0.75 | 19.8±0.40 | 13.0±0.63 |
> | + Consistency Fine-tuning| **68.6**±1.20 | **58.0**±1.10 | **37.4**±1.33 | **23.4**±1.20 | **17.4**±1.02 |
>
> Additionally, while our framework enhances the effectiveness of the one-step purification process, the underlying randomized smoothing method [1] still necessitates extensive sampling times N (e.g. $N=10,000$) for each test example evaluation. This extensive sampling is required to construct a reliable confidence interval for the certified robustness radius statistically. Furthermore, evaluations at various noise levels are required for each line of results presented in our Table 2. Consequently, evaluating the certified accuracy across the entire 50,000 test examples of ImageNet-64 remains a significant time cost.
>
> [1] Cohen et al. Certified adversarial robustness via randomized smoothing. international conference on machine learning. PMLR, 2019.
>
> > Why can only ensure generation is not enough for denoising?
>
> Even if the recovered sample is on-manifold, it can have a different semantic meaning from the original. The on-manifold property benefits generation but not denoising, where semantic consistency is crucial. For example, with a two-point data distribution at $\{-1, 1\}$ where $p(x=1)=p(x=-1)=0.5$, and the noisy distribution is similar, if a purification pipeline maps $-1$ to $1$ and $1$ to $-1$, the generation quality is perfect, but denoising fails.
>
> > Concerns about the importance of small transport
>
> As shown in lines 188-191 and Theorem 3.3, given an attack $\epsilon$ on a data point $x$, for a purification framework $d$, we obtain the recovered data $\hat{x} = d(x+\epsilon)$. The lower the transport from $\hat{x}$ to $x$, the more likely successful purification is. Figure 2 and Table 1 demonstrate that our purification framework $d$ reduces this transport compared to baselines through fine-tuning. Specifically, Figure 2 shows uniform transport reduction for recovered data across all $\sigma$ with CM, and consistency fine-tuning further reduces transport compared to the original model.
>
> We do not claim the transport between the data and the noisy distribution after the attack is small. If such transport is small, the attack is minimal, and successful purification is easier. Our theory applies to attacks causing large transport, such as Gaussian noise attacks.

---

> > ### Comment · Reviewer_wtha · 2024-08-08
> >
> > Thank you for your detailed response. Regarding the model evaluation using a subset of 500 images, it is better to use 512 images for evaluation to align with many diffusion purification settings ( "Diffusion Models for Adversarial Purification" and "Robust Evaluation of Diffusion-Based Adversarial Purification"). I appreciate your detailed explanation of the contribution, and the rating has been adjusted accordingly.

---

> > > ### Author Response · Authors · 2024-08-11
> > > **Response to your valuable feedback**
> > >
> > > Thank you for your response! We have expanded the 500 test images to 512 for evaluation to align with common diffusion purification settings ('Diffusion Models for Adversarial Purification' and 'Robust Evaluation of Diffusion-Based Adversarial Purification'). The results in the following table show that consistency purification with consistency fine-tuning, consistently achieves the highest certified accuracy, demonstrating the effectiveness of our approach.
> > >
> > > |                       | 0.0  | 0.05 | 0.15 | 0.25 | 0.35 |
> > > |----|----|----|----|----|----|
> > > | onestep-DDPM          | 55.3 | 44.7 | 33.4 | 15.2 |  8.8 |
> > > | Consistency Purification | 62.3 | 54.3 | 35.2 | 19.7 | 13.1 |
> > > | + Consistency Fine-tuning   | **68.8** | **58.0** | **37.5** | **23.4** | **17.4** |

---

### Official Review · Reviewer_43Xy · 2024-07-12

**Soundness:** 3
**Presentation:** 2
**Contribution:** 3
**Rating:** 5
**Confidence:** 3

**Summary:**

This paper introduces Consistency Purification, a novel framework that integrates consistency models with diffusion purification to enhance the certified robustness of deep neural networks, achieving efficient and effective image purification. The framework is further refined through Consistency Fine-tuning with LPIPS loss to ensure semantic alignment between the original and purified images, demonstrating state-of-the-art performance in both certified robustness and efficiency compared to existing methods.

**Strengths:**

The motivation is agreeable, and the paper is well-structured.
Theoretical explanations on the advantages of Consistency Purification are provided and clearly written.

**Weaknesses:**

It's noteworthy that there have been non-diffusion-based prior studies for enhancing certified robustness. A comparative analysis with those works would add depth to this study.
The experimental setting, including baseline selection and the selection of evaluation data, is not consistent with the baseline method One-Step DDPM. In this paper, only 100 samples from ImageNet are selected for evaluation. It would be beneficial to evaluate the method across multiple benchmarks and with more test samples.

**Questions:**

See Weakness.

---

> ### Author Rebuttal · Authors · 2024-08-07
>
> > Question A: Comparative analysis with non-diffusion-based pripor studies.
>
> To compare our consistency purification method with various non-diffusion-based approaches, we conducted additional experiments to compute the certified accuracy under three non-diffusion-based methods [1,2,3]. Cohen et al. [1] first proposed training a classifier with noisy images to ensure certified robustness. Subsequent works [2,3] build on Cohen et al.'s methodology, attempting to enhance the smoothed classifier by adding prediction consistency regularization, or incorporating per-sample bias.
>
> | | 0.0 | 0.25 | 0.50 | 0.75 | 1.00 |
>  |-------|------|------|------|------|------|
> | RS [1] | 74.8 | 59.2 | 42.0 | 31.8 | 22.0 |
> | Regularization [2] | 74.4 | 66.0 | 56.2 | 41.4 | 32.8 |
>  | ACES [3] | 74.6 | 66.4 | 57.0 | 43.6 | 32.8 |
> | Consistency Purification | **90.4** | 77.2 | 59.8 | 42.8 | 33.2 |
>  | + Consistency Finetune | 90.2 | **79.4** | **62.4** | **43.8** | **35.4** |
>
> The experimental results presented in the table show that our method surpasses all previous non-diffusion-based methods in achieving higher certified accuracy, particularly with a significantly high clean performance at $\sigma=0.0$.
>
> Furthermore, we would like to highlight that, in contrast to non-diffusion-based methods which incur significant costs by requiring additional fine-tuning of robust classifiers for each specific noise level, our method can be applied directly to any off-the-shelf classifiers. This significantly broadens its practical applications.
>
> [1] Jeremy Cohen, Elan Rosenfeld, and Zico Kolter. Certified adversarial robustness via randomized smoothing. In international conference on machine learning, pp. 1310-1320. PMLR, 2019.
>
> [2] Jongheon Jeong, and Jinwoo Shin. Consistency regularization for certified robustness of smoothed classifiers. Advances in Neural Information Processing Systems 33 (2020): 10558-10570.
>
> [3] Miklós Z. Horváth, Mark Niklas Müller, Marc Fischer, and Martin Vechev. Robust and Accurate--Compositional Architectures for Randomized Smoothing. arXiv preprint arXiv:2204.00487 (2022).
>
>
> > Question B: Experimental setting of evaluation data selection.
>
> All experiments in our paper utilize the same selection criteria for evaluation data. For the CIFAR-10 dataset, we selected 500 test examples from the 10,000 CIFAR-10 test set, choosing every 20th example in sequence (e.g., the 1st, 21st, 41st, etc.). Similarly, for the ImageNet-64 dataset, we selected 100 test examples from its 50,000  test examples using a fixed interval of 500. This consistent approach ensures that all evaluation datasets, including the baseline method One-Step DDPM, are identical across all experiments, thereby guaranteeing fair comparisons with our proposed method.
>
> > Question C: Limited evaluation examples for ImageNet-64 dataset.
>
> Here we include an additional experiment on ImageNet-64 using 500 samples, selecting every 100th example in sequence from the 50,000 ImageNet-64 test set. We present the certified accuracy of Consistency Purification in comparison with One-Step DDPM in the table below. The results consistently show that our method, with consistency fine-tuning, achieves the highest certified accuracy across the 500 test set of ImageNet-64, demonstrating the effectiveness of our approach.
>
> |                       | 0.0  | 0.05 | 0.15 | 0.25 | 0.35 |
> |-----------------------|------|------|------|------|------|
> | onestep-DDPM          | 55.2 | 44.8 | 33.4 | 15.2 |  8.8 |
> | Consistency Purification | 62.4 | 54.2 | 35.2 | 19.8 | 13.0 |
> | + Consistency Fine-tuning    | **68.6** | **58.0** | **37.4** | **23.4** | **17.4** |

---

> > ### Author Response · Authors · 2024-08-11
> > **Look forward to your reply**
> >
> > Dear Reviewer 43Xy,
> >
> > The deadline for the discussion period is approaching. We have provided our rebuttal material and hopefully could address your concerns. Your feedback is highly valuable to us, and we would greatly appreciate it if you could take some time to review our response.
> >
> > Best Regards,
> >
> > Authors

---

### Decision · Program_Chairs · 2024-09-25

**Decision:**

Accept (poster)

**Comment:**

This paper introduces Consistency Purification, a novel framework that integrates consistency models with diffusion purification to enhance the certified robustness of deep neural networks.  The paper provides both theoretical and experimental verification of the proposed method, and is well-written.  Some concerns raised by the Reviewers on the limited novelty and experimental evaluations were successfully addressed by the authors’ rebuttal.  The authors should incorporate all revision and improvement promised in the rebuttal into the final version of the paper.